# Nutritional and Chemical Quality of Maize Hybrids from Different FAO Maturity Groups Developed and Grown in Serbia

**DOI:** 10.3390/plants13010143

**Published:** 2024-01-04

**Authors:** Ivica Djalovic, Nada Grahovac, Zorica Stojanović, Ana Đurović, Dragan Živančev, Snežana Jakšić, Simona Jaćimović, Caihuan Tian, P. V. Vara Prasad

**Affiliations:** 1Institute of Field and Vegetable Crops, National Institute of the Republic of Serbia, 21000 Novi Sad, Serbia; maizescience@yahoo.com (I.D.); dragan.zivancev@ifvcns.ns.ac.rs (D.Ž.); snezana.jaksic@ifvcns.ns.ac.rs (S.J.); simona.jacimovic@ifvcns.ns.ac.rs (S.J.); 2Faculty of Technology Novi Sad, University of Novi Sad, Bulevar cara Lazara 1, 21000 Novi Sad, Serbia; zorica.stojanovic@uns.ac.rs (Z.S.); ana.djurovic@uns.ac.rs (A.Đ.); 3State Key Laboratory of Vegetable Biobreeding, Institute of Vegetables and Flowers, Chinese Academy of Agricultural Sciences, Beijing 100081, China; tiancaihuan@caas.cn; 4Department of Agronomy, Kansas State University, Manhattan, KS 66506, USA; vara@ksu.edu

**Keywords:** maize, FAO maturity groups, nutritional composition, antioxidant properties, fatty acid composition

## Abstract

Maize is a globally significant cereal crop, contributing to the production of essential food products and serving as a pivotal resource for diverse industrial applications. This study investigated the proximate analysis of maize hybrids from different FAO maturity groups in Serbia, exploring variations in polyphenols, flavonoids, carotenoids, tocopherols, and fatty acids with the aim of understanding how agroecological conditions influence the nutritional potential of maize hybrids. The results indicate substantial variations in nutritional composition and antioxidant properties among different maturity groups. The levels of total polyphenols varied among FAO groups, indicating that specific hybrids may offer greater health benefits. Flavonoids and carotenoids also showed considerable variation, with implications for nutritional quality. Tocopherol content varied significantly, emphasizing the diversity in antioxidant capacity. Fatty acid analysis revealed high levels of unsaturated fatty acids, particularly linoleic acid, indicating favorable nutritional and industrial properties. The study highlights the importance of considering maturity groups in assessing the nutritional potential of maize hybrids.

## 1. Introduction

Maize (*Zea mays* L.) is, along with rice and wheat, one of the most widespread cereal crops in the world. The total global production of maize in 2021 was above 1200 million tons, with both the Americas and Asia contributing more than 48% and 31%, respectively. Europe was ranked fourth among all continents in annual production, totalling 142 million tons, with the Balkan region countries (Bosnia and Herzegovina, Bulgaria, Croatia, Hungary, Romania, and Serbia) contributing approximately about 24% [1]. Maize is utilized to produce a variety of food and processed products, including starch, sweeteners, oils, beverages, adhesives, industrial alcohol, and fuel ethanol [2]. In human nutrition, maize, along with the two aforementioned cereals, accounts for at least 30% of food calories in 94 countries, benefiting over 4.5 billion people [3]. In light of prevailing climate change dynamics, compounded by escalating drought instances and amplified evapotranspiration rates significantly affecting maize susceptibility, the imperative to discern hybrids exhibiting augmented biological nutritional worth coupled with precise alignment to pertinent FAO maturity groups is markedly heightened [4,5].

Maize, from a nutritional standpoint, boasts a noteworthy chemical composition that distinguishes it among cereals. Typically, the oil content in maize kernels ranges from 2% to 6%, with approximately 85% primarily concentrated in the germ’s scutellum [6]. While oil constitutes a minor fraction of the grain, its quantity can differ among maize varieties with distinct genetic backgrounds [7]. The lipids in maize can be categorized as nonpolar, polar, and non-saponifiable [8]. The most prevalent type is the nonpolar fraction, which includes fatty acids (FA) and triglycerides rich in linoleic acid (18:2), followed by oleic acid (18:1) (OFA). Linoleic acid (LFA) is recognized as the singular essential fatty acid for humans, and maize oil makes up over 50% of this specific FA. About 99% of maize oil’s composition comprises fatty acids such as palmitic (16:0), stearic (18:0), oleic (18:1), linoleic (18:2), and linolenic (18:3) acids [9]. Among these, monounsaturated oleic acid (OLA) constitutes 24%, and polyunsaturated linoleic acid (LOA) constitutes 62% [6]. Consuming crude or refined corn oils rich in linoleic acid (LOA) enhances the balance between saturated and unsaturated dietary fats. Regrettably, it does not impact the ratio between omega-6 and omega-3 fatty acids (FA). Linoleic acid (18:3) can undergo conversion in the human system to linolenic acid (LLA) and subsequently to other significant long-chain fatty acids and metabolites, including eicosanoids and prostaglandins [6]. From a nutraceutical perspective, polyunsaturated omega-3 fatty acids are acknowledged for their ability to decrease serum cholesterol. The percentages of unsaturated fatty acids (UFA), particularly oleic and linoleic acids, in maize oil are influenced by various factors, including maize variety, agroclimatic conditions, geographical location, agronomic factors and harvesting and processing methods. Respectively, the percentages of unsaturated fatty acids in maize oil are a result of complex interactions between the genetic makeup of the maize variety, the environmental conditions in which it is grown, and various agricultural and processing practices. 

The protein content in maize, a crucial nutritional component, usually hovers around 10%, providing a valuable source of dietary protein. Corn oil is acknowledged as an exceptional source of tocopherols, which serve as antioxidants and contribute significantly to vitamin E intake. Similar to essential fatty acids, vitamin E is a vital component of the human diet, as the body cannot synthesize it internally. Tocopherols, a robust class of antioxidants, play a crucial role in safeguarding polyunsaturated fatty acids within membranes from degradation caused by reactive oxygen species like ozone, singlet oxygen, peroxides, and hyperoxides [10]. The antioxidant function of tocopherols is not only pivotal for health, but also essential for maintaining oil quality, extending its shelf life by inhibiting the onset of rancidity. Corn oil typically contains four major tocopherols: alpha-, beta-, gamma-, and delta-tocopherol. Among these, gamma-tocopherol predominates in commercially available corn oil, followed by alpha-tocopherol and delta-tocopherol. Delta-tocopherol, in particular, exhibits the most potent antioxidant effect, while alpha-tocopherol boasts the highest vitamin E activity. 

Maize displays a diverse range of colors, spanning from white to yellow, red, blue, purple, and more. Kernels with blue, purple, and red hues are particularly abundant in anthocyanins, known for their well-established antioxidant and bioactive properties [11]. The content of anthocyanin, carotenoid, and phenolic compounds in maize varies according to its coloration. The pericarp portion holds the highest concentration of anthocyanin pigments, while the aleurone layer contains smaller amounts [12]. Maize exhibits notable natural variability in kernel carotenoids, with certain genotypes accumulating as much as 66.0 μg g^−1^ [13]. Floury maize tends to have fewer carotenoids compared to yellow maize. Typically, provitamin A carotenoids make up only 10–20% of the total carotenoid content in maize, while zeaxanthin and lutein each represent 30–50%. In standard maize varieties, concentrations of provitamin A carotenoids, namely α-carotene, β-carotene, and β-cryptoxanthin, range from 0 to 1.3, 0.13 to 2.7, and 0.13 to 1.9 nmol g^−1^, respectively [14]. In its natural state, maize serves as an abundant reservoir of carotenoids, including beta-carotene, zeaxanthin, lutein, and cryptoxanthin. These carotenoids offer a wide array of health benefits, spanning from the preservation of normal vision to the reduction of oxidative stress.

Maize is recognized for its rich content of various phenolic acids, with ferulic acid standing out as a crucial phytochemical whose concentration varies among different maize types. High-carotenoid maize, in particular, exhibits a higher total ferulic acid content compared to white, yellow, red, and blue maize varieties. It is noteworthy that a significant portion of the ferulic acid in maize exists in a bound form [11]. The majority of phenolics, including phenolic acids, flavonoids and conjugated amines, are predominantly concentrated in the pericarp and aleurone layers, as well as the germ, with minimal traces found in the endosperm [15]. Total polyphenols, found in maize kernels, possess antidiabetic, anti-obesity, and anticancer effects, contributing to the preservation of the human cardiovascular system.

The chemical composition of maize, especially in hybrid varieties studied in this research, is subject to variations influenced by genetic factors, agroclimatic conditions, and specific breeding practices. Understanding these factors is crucial for elucidating the nutritional potential and health benefits associated with consuming different maize hybrids.

The purpose of this research is to investigate and understand the proximate analysis of maize hybrids from different FAO maturity groups in Serbia. The results of the study reveal substantial variations in the nutritional composition and antioxidant properties among maize hybrids from different FAO maturity groups. The levels of total polyphenols, flavonoids, carotenoids, tocopherols, and fatty acids varied significantly, indicating that specific hybrids within different maturity groups may offer distinct health benefits. The emphasis on maturity groups underscores the importance of considering this factor when assessing the nutritional potential of maize hybrids. Overall, the research provides valuable insights into the diversity of nutritional content in maize hybrids, with implications for both food production and industrial applications.

## 2. Materials and Methods

### 2.1. Materials

A set of 30 NS hybrids, an FAO 100–700 maturity group, were grown in a randomized complete block design (RCBD) at the Institute of Field and Vegetable Crops, Novi Sad, Serbia in 2021 (45°20′14″ N, 19°51′44″ E, 78 m above sea level). Maize hybrids of standard grain quality with different vegetation lengths and production purposes were grown, as well as hybrids with specific properties. Three of them belonged to the FAO 100–200 maturity group (G11, G24, and G27 hybrids), seven to the FAO 200–400 maturity group (G4, G6, G8, G10, G15, G19, and G26 hybrids), nine to the FAO 400–600 maturity group (G2, G5, G7, G9, G17, G20, G22, G28, and G30 hybrids), and eleven to the FAO 600–700 maturity group (G1, G3, G12, G13, G14, G16, G18, G21, G23, G25, and G29 hybrids). The experiment was carried out on a chernozem soil, with a humus content of 2.86%, pH = 7.04, total nitrogen 0.24%, P_2_O_5_ 27.53 mg 100 g^−1^, and K_2_O 29.23 mg 100 g^−1^. The preceding crop was soybean in a 3-year rotation (soybean–maize–wheat). Winter wheat (*Triticum aestivum* L.) was the previous crop. The selected plots underwent plowing in October to a depth of up to 30 cm, followed by seedbed preparation before sowing in March using heavy-duty cultivators (multi-tiller) at a depth of 15 cm. On 22 April 2021, the crop was sown using a Wintersteiger AG pneumatic precision seed drill at a depth of 5 cm. The plot dimensions were 5 × 2.8 m, featuring intra-row spacing of 22 cm and row spacing of 70 cm. Throughout both years, weed control was executed through conventional chemical methods. Pre-emergence application included a dose of 1.4 l ha^−1^ combined with 3.5 L ha^−1^ of a mixture containing 375 g L^−1^ S-Metolachlor, 125 g L^−1^ Terbuthylazine and 37.5 g L^−1^ Mesotrione. Post-emergence application was also conducted. During the vegetation season, control of *Sorghum halepense* sp. and other narrow-leaved weeds was achieved by applying Nicosulfuron or Rimsulfuron at a rate of 50–60 g ha^−1^. All maize hybrids are classified into international FAO maturity groups, depending on the length of the growing season (Table 1). The FAO group 100 is the earliest, and 700 the latest maturity group. A higher number indicates a longer growing season and whether more heat units are required for the variety to reach grain maturity.

Serbia experiences a temperate continental climate, characterized by distinctive local features and a gradual shift between seasons. The maize crop yields are significantly influenced by weather factors, particularly precipitation and temperature patterns. Meteorological data, encompassing temperature and total precipitation, were gathered from the automated weather station at the Rimski Sancevi Agrometeorological Experimental Station [17]. Novi Sad (Rimski Sancevi), situated in northern Serbia at coordinates 45°20′ N, 19°51′ E, standing on the Pannonian Plain at an elevation of 80–86 m above sea level.

### 2.2. Chemicals and Reagents

All chemicals utilized in this study were of analytical reagent grade. Quercetin, gallic acid, β-carotene, and DPPH (1,1-diphenyl-2-picrylhydrazyl) were acquired from Sigma-Aldrich (St. Louis, MA, USA). Folin–Ciocalteu reagent was supplied by AppliChem (Darmstadt, Germany). Sodium sulfate anhydrous, sodium carbonate, sodium hydroxide, and aluminum chloride were purchased from Carl Roth (Karlsruhe, Germany). Methanol, acetone, and petroleum ether were sourced from VWR Chemicals BDH (Fontenay-sous-Bois, France), while sodium nitrite and ascorbic acid were obtained from Centrohem (Stara Pazova, Serbia). Double-distilled water was used throughout all experiments.

### 2.3. Samples and Sample Preparation

Prior to analysis, the samples were homogenized using an IKA MultiDrive basic mill (IKA, Staufen, Germany), sieved using a 60-mesh sieve, and then subjected to different extraction protocols depending on the types of analytes to be analyzed. Total antioxidants were extracted and quantified according to [18]. Briefly, 1 g of grounded maize sample was mixed with 80% methanol in an ultrasound bath (Iskra, Kranj, Slovenia) for 30 min. Subsequently, the mixture was subjected to agitation on a magnetic stirrer (HedaS, Vršac, Serbia) for a duration of 24 h at room temperature (25 ± 1 °C). The obtained extracts were placed in a centrifuge (Ultra-8V, LW Scientific, Lawrenceville, NJ, USA) and centrifuged at 6000 rpm for 5 min, and the resulting supernatant was then filtered using a 45 µm filter from Macherey-Nagel (Düren, Germany). During the whole procedure, extracts were protected from light by covering conical flasks with aluminum foil. The crude methanolic extracts were utilized for subsequent analysis of total phenolics, total flavonoids, and DPPH antioxidant activity. The carotenoids were extracted from maize samples following the protocol as described by Hossain and Jayadeep [19] with some modifications. Around 2 g of milled sample was mixed with 10 mL extraction solvent (acetone–petroleum ether with a boiling point of 40–60 °C in a 1:1 ratio (*v*/*v*)) with a magnetic stirrer (HedaS, Vršac, Serbia) for 45 min. After centrifugation (Ultra-8V, LW Scientific, Lawrenceville, NJ, USA) at 6000 rpm for 10 min at room temperature, the supernatant was collected and extraction was repeated until the extraction solvent became colorless. Subsequently, acetone was washed from the collected extract using a sufficient amount of double-distilled water. Afterward, the petroleum ether’s carotenoid extract was transferred to a volumetric flask through the anhydrous sodium sulfate in order to remove the residual water, and the flask was filled up to mark with the same solvent. During the whole sample preparation procedures, all laboratory glassware were coated with aluminum foil to ensure protection from light. Obtained extracts were used for subsequent spectrophotometric analysis of total carotenoids.

### 2.4. Chemical Analysis

The moisture and protein content of the maize samples were determined by the standard analytical methods of the American Association of Cereal Chemists (AACC) numbers 44-15.02 and 46-16.01 [20], whereas oil content was determined by to the Association of Official Agricultural Chemists (AOAC) method No. 920.85 [21], using petroleum ether (boiling point 40–65 °C) (Fisher Scientific, Fair Lawn, NJ, USA) as the solvent in Soxhlet apparatus (Soxtherm 2000 automatic, Gerhardt, Germany), following the manufacturer’s instructions. The determination of ash content was performed according to the ISO standard method number 749 [22]. The moisture, protein, oil and ash contents of samples were expressed in g 100 g^−1^ of maize seeds. The energy values were calculated by multiplying the mean values of fats, proteins and total carbohydrates using At water factors of 37 kJ g^−1^ (9.0 kcal g^−1^), 17 kJ g^−1^ (4.0 kcal g^−1^), and 17 kJ g^−1^ (4.0 kcal g^−1^), respectively [23]. The results are expressed as kJ 100 g^−1^.

### 2.5. Antioxidant Properties and Carotenoids

#### 2.5.1. Total Antioxidant Quantification

To assess the antioxidant properties of the maize samples being studied, the total phenolic and flavonoid contents were estimated, along with the DPPH radical scavenging assay. The total phenolic content was analyzed using the Folin–Ciocalteu spectrophotometric method [18] with a slight modification. Briefly, in a 10 mL volumetric flask, 0.5 mL of the maize extract was mixed with 6.0 mL of double-distilled water and 0.5 mL of Folin–Ciocalteu reagent for 60 s. Subsequently, 2.0 mL of 15% sodium carbonate solution was added to the mixture, and the solution was mixed for 30 s. Then, double-distilled water was used to bring the total volume to 10 mL. The mixture was incubated for 45 min in the dark, and the absorbance of the resulting blue color solution was measured at 760 nm using a UV–Vis spectrophotometer (UV-2100, Unico Instrument Co., Shanghai, China). Quantification was performed with a standard calibration curve using gallic acid, and the total phenolic content was expressed as milligrams of gallic acid equivalent per kg of sample dry weight (mg GAE kg^−1^ DM). For the quantification of the total flavonoids in maize, the aluminum chloride spectrophotometric method was used [24]. Then, 1.0 mL of the maize extract was added to a 10 mL volumetric flask containing 4.0 mL double-distilled water. Then, 0.3 mL of 5% sodium nitrite solution was added. Five minutes later, 0.3 mL of 10% aluminum chloride solution was added. After 1 min, 2.0 mL of 1 mol L^−1^ sodium hydroxide solution was added, and the final volume was set to 10 mL by the addition of double-distilled water. Following a 15 min incubation period in a dark environment at room temperature, absorbance was recorded at a wavelength of 510 nm. To establish the calibration curve, standard solutions of quercetin were employed, and the total flavonoid content was expressed as milligrams of quercetin equivalent per kilogram of dry matter (mg QE kg^−1^ DM). A 2,2-diphenyl-1-picrylhydrazyl (DPPH) free radical assay was employed to evaluate the potential antioxidant activity of the maize extracts. The DPPH assay was conducted following a previously described method [24] with some modification. In this assay, 0.5 mL of extract was combined with 4.0 mL of double-distilled water and 2.5 mL of DPPH ethanolic solution. The mixture was incubated at room temperature for 15 min. Subsequently, the changes in absorbance at 517 nm were measured, and the antioxidant activity was determined by calculating the percentage inhibition resulting from hydrogen donor activity. Antioxidant capacities were expressed as milligrams of ascorbic acid equivalent per gram of dry matter (mg AAE g^−1^ DM), based on a standard curve generated using ascorbic acid solutions.

Measurements were conducted in triplicate, and the results are reported as the mean ± standard deviation.

#### 2.5.2. Estimation of Total Carotenoid Content

The total carotenoid contents in obtained extracts were estimated using the spectrophotometric method [25]. The absorbance of the prepared extracts was measured at 450 nm against petroleum ether using a UV–Vis spectrophotometer (UV-2100, Unico Instrument Co., Shanghai, China) in a 1 cm cell. β-carotene was utilized as a reference compound for quantification through a standard calibration curve. The total carotenoid amount was expressed as mg β-carotene equivalent per kg of dry sample (mg β-CE kg^−1^ DM). All measurements were performed in triplicate, and results are presented as mean ± standard deviation.

### 2.6. Lipids, Fatty Acids and Tocopherols’ Analysis

#### 2.6.1. Lipid Extraction and Fatty Acid Analysis

The maize seeds were ground using an IKA mill (A11 basic, IKA-Werke GmbH & Co. KG, Staufen, Germany) to prepare powdered samples. The seed powder was placed in a petroleum ether solvent to extract oil from the powder for 8 h with Soxhlet apparatus using an automatic solvent extraction system (60 °C). After the extraction procedure, the triacylglycerols of maize oil samples were chemically converted into their volatile fatty acid methyl esters (FAMEs). FAMEs were prepared according to AOCS official method No. Ce 2-66 [26], with some modifications according to Purar et al. [27]. The fatty acid composition of extracted maize oil was determined by injecting the aliquot of 1 µL (split ratio 1:70) into gas chromatography (Konik-Tech HRGC 4000 SA, Barcelona, Spain) equipped with a flame ionization detector and Omegawax 250 column (30 m length, 0.25 mm ID and 0.25 µm film thickness) using helium as the carrier gas with a constant flow rate of 1 mL min^−1^. The temperatures of the detector and injector were maintained at 250 °C. The FAMEs were separated using a temperature gradient program [27]. FAMEs were identified based on pure FAME standard mixture (Supelco, Bellefonte, PA, USA, FAME, RM-1) by comparing GC retention times and the Kovats retention index with reference analytical standards. The analysis was repeated twice with two replicates under the same conditions. The result of the individual FA content was expressed as g 100 g^−1^ of oil, representing the percentage composition (%).

#### 2.6.2. Tocopherol Analysis

The tocopherol content in the maize oil was assessed using a Sykam (GmbH, Kleinostheim, Germany) high-performance liquid chromatograph (HPLC) following the standard method of AOCS [28] with a minor modification. A 300 µL aliquot of maize oil was placed into 2 mL volumetric flasks, and n-hexane (Fisher Scientific, Fair Lawn, NJ, USA) was added, swirling to dissolve the oil sample and making up a volume with the same solvent. An aliquot of 1 mL of this solution was filtered through a regenerated cellulose filter (0.22 µm, Macherey Nagel) and transferred into the vial for further HPLC analysis. The amount of tocopherols in the obtained samples was determined using a fluorescence detector (excitation wavelength 280 nm and emission wavelength 340 nm). A 20 µL aliquot of sample was injected on a Nucleosil 100-5 NH_2_ column (25 cm × 4.6 mm, particle size 5 μm, pore size 100 Å, Macherey Nagel). The sample was isocratically eluted with 30% (*v*/*v*) ethyl acetate in HPLC-grade n-hexane at a 1 mL min^−1^ flow rate. The column oven was maintained at 40 °C. A calibration curve was prepared with mixture standards of tocopherols (α-, β-, γ-, δ-tocopherol) provided by Sigma Aldrich (St. Louis, MO, USA). All the analyses were performed twice.

### 2.7. Statistical Analysis

Analytical measurements of the basic chemical composition, antioxidant properties and carotenoids were performed in triplicate, whereas tocopherols and fatty acid analysis were performed in duplicate. The data were statistically analyzed by a one-way factorial analysis of variance (ANOVA), followed by the comparison of mean values based on Duncan’s multiple means comparison tests with a 95% confidence level. The ANOVA analysis was performed by InfoStat software Version 2016e (UNC, Córdoba, Argentina), whereas the PCA analysis was performed by XLSTAT-Pro, demo version, Version 5.03, 2014 software (Addinsoft, Paris, France).

## 3. Results and Discussion

### Maize Chemical Composition

To systematically characterize maize hybrids from Serbia according to FAO maturity groups, we first conducted a proximate chemical composition analysis. The basic proximate chemical composition of maize hybrids within various FAO maturity groups is presented in Table 2. As expected, the predominant components of maize grains are carbohydrates, followed by proteins and fats. Maize hybrids in the FAO 400–600 and 600–700 maturity groups exhibited statistically higher carbohydrate contents than maize hybrids from the FAO 100–200 maturity group. The value close to 5% suggests that maize hybrids from the FAO maturity group could serve as interesting raw material for the oil industry [29]. Additionally, maize hybrids in the FAO 200–400 maturity group displayed the highest crude fat content, which was statistically higher than that of maize hybrids in the FAO 100–200 maturity group. These results align with the results obtained by [30] when they analyzed maize genotypes with different hardnesses and colors.

The analysis of maize hybrids within different FAO maturity groups provides valuable insights into their nutritional composition and antioxidant properties, as outlined in Table 3. The obtained data are crucial for assessing the potential health benefits of these hybrids and their suitability for various agricultural and dietary purposes.

The total polyphenol content in maize hybrids varies significantly among the different FAO maturity groups. The highest total polyphenol content was observed in the G1 hybrid of the FAO 600–700 maturity group, with a content of 1720.14 mg GA eq kg^−1^. On the other hand, the G30 hybrid of the FAO 400–600 maturity group had the lowest polyphenol content, measuring 711.86 mg GA eq kg^−1^. The variation in total polyphenol content implies that certain hybrids could provide more substantial health benefits, owing to their elevated antioxidant capacity. Flavonoids, a subclass of polyphenols, also exhibited substantial variation across different hybrids. The G3 hybrid of the FAO 600–700 maturity group showed the highest flavonoid content at 1577.59 mg Q eq kg^−1^, while the G30 hybrid of the FAO 400–600 maturity group had the lowest content at 686.61 mg Q eq kg^−1^. Recognized for their antioxidant properties and potential health benefits, higher flavonoid content in hybrids makes them more attractive from a nutritional perspective. Antioxidant activity was assessed using the DPPH assay, and the results indicated varying levels of antioxidant potential. The G1 hybrid of the FAO 600–700 maturity group exhibited the highest antioxidant activity at 0.38 mg AA eq g^−1^ dry matter, while the G30 hybrid of the FAO 400–600 maturity group had the lowest activity at 0.07 mg AA eq g^−1^ dry matter. These differences in antioxidant activity are in accordance with variations in polyphenol and flavonoid contents. The range of total polyphenol content in our study was found to be comparable to the results reported by Ranilla et al. [31], mainly ranging from 1331.4 to 1703.9 mg GA eq kg^−1^, with one exceptional sample containing 5488.7 mg GA eq kg^−1^. However, Fuentes-Cardenas et al. [32] observed slightly higher total polyphenol contents in their analyzed maize samples, which ranged from 2015 to 2911 mg GA eq kg^−1^. These variations in total polyphenol content can be attributed to several factors, including differences in maize hybrids, cultivation, and harvesting practices, as well as disparities in sampling, sample preparation, and analytical procedures.

Carotenoids, including β-carotene, are important pigments found in maize and are known for their role in promoting eye health and overall immunity. The G22 hybrid of the FAO 400–600 maturity group displayed the highest carotenoid content at 98.15 mg β-C eq kg^−1^, while the G16 hybrid of the FAO 600–700 maturity group had the lowest content at 12.85 mg β-C eq kg^−1^. However, the majority of the analyzed maize hybrids contained carotenoid content above 58 mg kg^−1^. In comparison to previously published data [19,33], our results suggest comparable but also higher total carotenoid contents.

The total tocopherol contents of tested maize hybrids are presented in Table 4. Among the various FAO maturity groups, there is substantial variation in the total tocopherol content of maize hybrids. The G1 hybrid in the FAO 600–700 maturity group exhibited the highest total tocopherol content, measuring 701.11 mg kg^−1^. Conversely, the G4 hybrid in the FAO 200–400 maturity group displayed the least total tocopherol content, registering at 19.60 mg kg^−1^. As minor constituents, tocopherols have a substantial positive effect on the nutritional and/or technological properties of the oil [34]. Nutritionally, tocopherols, along with other related compounds (tocotrienols), possess vitamin E activity. They act to inhibit oxidative, proliferative, metabolic, and inflammatory damage at the cellular level, thereby preventing the onset of various chronic diseases [35]. In food applications like frying oil, fried snacks, and margarine, tocopherols play the role of antioxidants [36]. The diversity in overall tocopherol content implies that specific hybrids could provide more significant health benefits, owing to their elevated antioxidant capacity. The G1 hybrid of the FAO 600–700 maturity group exhibited the highest α-tocopherol content at 205.42 mg kg^−1^. In the same FAO group, the highest γ-tocopherol content was recorded in the G1 hybrid at 478.71 mg kg^−1^, while the lowest was observed in the G12 hybrid at 5.15 mg kg^−1^. The G7 hybrid (FAO 400–600 maturity group) exhibited the highest values for β- and δ- tocopherols, measuring 28.96 mg kg^−1^ and 30.28 mg kg^−1^, respectively. Technologically, tocopherols serve to protect the oil from oxidative deterioration, primarily targeting the double bonds of unsaturated fatty acids [37]. At moderate temperatures and low tocopherol concentrations, α-tocopherol typically exhibits superior antioxidant properties compared to γ-tocopherol, but the scenario reverses at high tocopherol concentrations [38,39]. The total tocopherol content in our study was in line with the findings of other authors [40,41], primarily ranging from 39.9 to 1238.17 mg kg^−1^. Compared to previously published findings [41,42,43], our results indicate comparable yet slightly elevated α- and γ-tocopherol levels.

Statistical analysis indicates substantial variations in antioxidant properties observed among the different maize hybrids. Our comprehensive results reveal that maize hybrids within the same FAO group can exhibit significant variations in nutritional content. For example, in the FAO 600–700 group, we observed a wide range of values for most parameters, suggesting that selecting specific hybrids within the same FAO group can lead to substantial differences in nutritional quality. The fluctuations in overall antioxidant and bioactive compound contents and tocopherol content can be ascribed to various factors, encompassing distinctions in maize hybrids, cultivation, and harvesting techniques, along with variations in sampling, sample preparation, and analytical methodologies.

While a higher oil content is a sought-after characteristic in maize due to its ability to offer increased energy to consumers, the quality of the oil, particularly concerning its fatty acid composition, plays a more significant role in human nutrition. The nutritional and industrial properties of an oil are influenced by its fatty acid composition, which can also impact its commercial value [44]. Table 4 presents the results obtained for fatty acid compositions. In general, the investigated oils exhibited high levels of UFA and low levels of saturated fatty acids (SFA), with mean values of 87.09% and 12.90%, respectively. This composition is deemed favorable from a health perspective. Within the UFA, polyunsaturated linoleic acid (C18:2) constituted the predominant component, with content from 49.68% to 52.17%, followed by monounsaturated oleic acid (C18:1c, 32.51–34.78%). Among the SFA, palmitic acid had the highest content of about 11%, while the contents of myristic, stearic, γ-linolenic, arachidic, eicosenoic, behenic, and lignoceric acids were under 2% (Table 5).

Among the maize hybrids in the FAO 100–200 maturity group, the investigated oil showed the highest levels of linoleic (52.17%), oleic (34.78%), and palmitic acids (11.81%). Conversely, the FAO 600–700 maturity group exhibited the lowest content of both linoleic (49.68%) and oleic acids (32.51%), while the lowest palmitic content (10.84%) was recorded in the FAO 400–600 maturity group. The findings presented by other researchers align with our results. Saini et al. [45] found that linoleic acid was the dominant fatty acid in maize oil, with a content from 38% to 58.9%, while the amounts of oleic acid and palmitic acid were 23.5–45.3% and 10.8–17.30%, respectively [45].

From a nutritional perspective, the emphasis extends beyond the content of individual polyunsaturated fatty acids to the crucial balance between n-6 and n-3 acids, each offering distinct health benefits. Adhering to nutritional guidelines, the recommended ratio should ideally range between 1:1 and 4:1 [18,46]. In this context, the maize oils under investigation, with n-6/n-3 ratios from 35:1 to 24:1, do not have the optimal ratio. The results indicate variations in the n-6/n-3 ratios among the tested hybrids, underscoring the significant genetic potential for improvement. The abundant presence of crucial linoleic acid (50–52%) enhances the nutritional value of maize seed oils for human consumption. This polyunsaturated fatty acid is vital for normal growth, health promotion, and the prevention of conditions such as coronary heart disease, atherosclerosis, and high blood pressure [47]. While beneficial for human health, oils rich in PUFA are susceptible to oxidation, resulting in instability and a limited shelf life, making them unsuitable for cooking or frying [48].

To understand how the basic chemical composition, antioxidant activity, and the contents of carotenoids and tocopherols affect maize hybrids of different FAO maturity groups, the results were subjected to a robust compositional PCA. The first principal component (PC1) accounted for over 25% of the variability, while the second principal component (PC2) explained nearly 20% of the variability (Figure 1). The content of crude fats exhibited a positive correlation with the content of total polyphenol, flavonoids, DPPH, SFA and β-tocopherols, while it showed a negative correlation with UFA. Total tocopherols were positively correlated with three forms of tocopherols (α-, γ-, and δ-), and negatively correlated with carotenoid content. All forms of tocopherols, except for β-tocopherols, were unrelated to content of carbohydrates and PUFA. These two parameters were negatively correlated with content of protein, ash and MUFA. The positive correlation between the total content of polyphenols, flavonoids and DPPH with ash explains that most of these compounds originate from the outer parts of the maize grain. Two maize hybrids from the FAO maturity group 200–400 (G6 and G8 hybrids) and four that belong to the FAO maturity group 600–700 (G1, G13, G14, and G25 hybrids) were closely associated with most tocopherols. On the other hand, three maize hybrids that belong to the FAO maturity group 600–700 (G12, G18, and G21 hybrids) and two maize hybrids from the FAO maturity group 100–200 (G11 and G24 hybrids) were closely linked to carotenoid content. According to the PCA biplot, maize hybrids with high protein and MUFA contents included G2, G4, G19, and G27 hybrids from different FAO maturity groups were closely associated with these parameters. Additionally, three maize hybrids from the FAO maturity group 400–600 (G5, G7 and G9 hybrids), and G3 hybrid that belong to group 600–700 were in close proximity to the parameters related to antioxidative activity (total polyphenols, flavonoids, DPPH, and β- tocopherols) and crude fat content. Conversely, according to the PCA biplot, maize hybrids with high carbohydrate and PUFA contents were observed in four hybrids from the FAO 600–700 maturity group (G10, G16, G23, and G29 hybrids) and two maize hybrids in the FAO 400–600 maturity group (G20 and G22 hybrids). Moreover, were noticed three hybrids from the FAO 400–600 maturity group (G17, G28 and G30 hybrids) and one hybrid from the FAO 200–400 maturity group (G26 hybrid) with high MUFA content.

## 4. Conclusions

This study provides valuable insights into the nutritional composition of maize hybrids across different FAO maturity groups in Serbia. The results highlight the significance of maturity group classification in elucidating variations in polyphenols, carotenoids, tocopherols, and fatty acids. Total polyphenol content ranged from 711.86 to 1720.14 mg GA eq kg^−1^, while total flavonoids varied from 686.61 to 1577.59 mg Q eq kg^−1^. Carotenoid content ranged from 12.85 to 98.15 mg β-C eq kg^−1^. The analysis revealed a diverse tocopherol profile among maize hybrids, with total tocopherol content ranging from 19.60 to 701.11 mg kg^−1^. The observed diversity in bioactive compound profiles among specific maize hybrids suggests the potential for these hybrids to offer significant health advantages and enhanced technological properties, attributable to their heightened antioxidant capacity. The prevalence of unsaturated fatty acids, particularly linoleic acid, positions these maize hybrids favorably for both nutritional and industrial applications. These findings contribute to our understanding of maize nutritional quality and offer insights for future selection of maize hybrids and cultivation practices to optimize the health-promoting components of this vital cereal crop.

## Figures and Tables

**Figure 1 plants-13-00143-f001:**
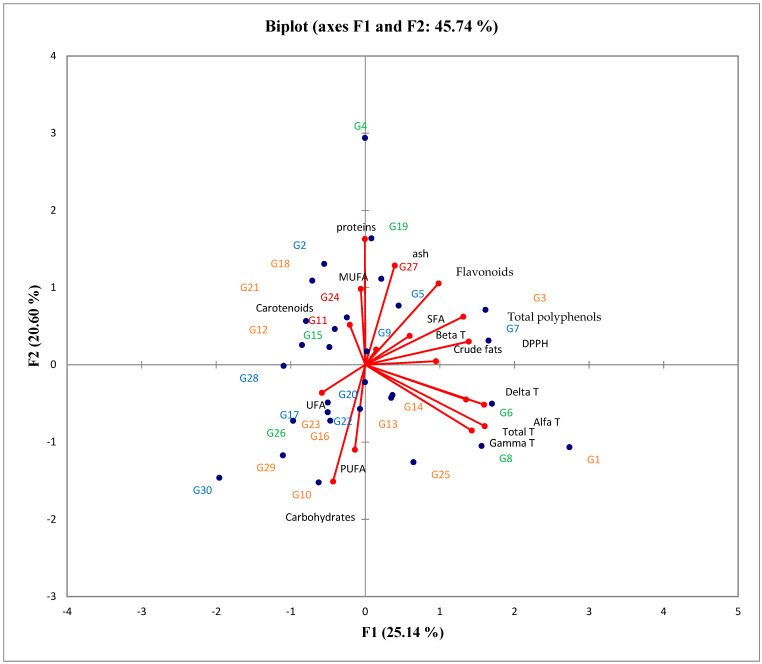
PCA biplot of the basic chemical composition, antioxidant activity and carotenoid and tocopherol contents of the examined maize hybrids belonging to different FAO groups, where 
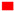
 marks the FAO 100–200 maturity group, 
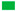
 marks the FAO 200–400 maturity group, 
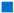
 marks the FAO 400–600 maturity group, and 
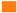
 marks the FAO 600–700 maturity group (alfa T-α-tocopherols, beta T-β-tocopherols, delta T-δ-tocopherols, gamma T-γ-tocopherols, total T-total tocopherols).

**Table 1 plants-13-00143-t001:** FAO maturity groups and the length of the maize growing period [16].

FAO Maturity Group	Length of the Growing Season (Number of Days)
FAO 100	95–100
FAO 200	100–105
FAO 300	105–110
FAO 400	110–115
FAO 500	115–120
FAO 600	120–125
FAO 700	130–145

**Table 2 plants-13-00143-t002:** The basic chemical composition of maize hybrids across the studied FAO maturity groups.

FAO Maturity Groups	Proximate Analysis (g 100 g^−1^) ^1^					Energy ^2^ (KJ 100 g^−1^)
	Moisture	Ash	Proteins	Crude Fats	Carbohydrates	
100–200	11.72 ± 0.16 ^A^	0.17 ± 0.01 ^A^	9.23 ± 0.39 ^A^	4.56 ± 0.86 ^AB^	74.32 ± 0.85 ^B^	1589.1
200–400	11.41 ± 0.40 ^B^	0.14 ± 0.02 ^B^	8.74 ± 1.33 ^A^	4.79 ± 0.82 ^A^	74.92 ± 1.41 ^AB^	1599.5
400–600	11.55 ± 0.27 ^AB^	0.15 ± 0.01 ^B^	8.74 ± 0.80 ^A^	4.47 ± 0.62 ^AB^	75.10 ± 0.78 ^A^	1590.7
600–700	11.65 ± 0.28 ^A^	0.14 ± 0.01 ^B^	8.74 ± 0.63 ^A^	4.22± 0.52 ^B^	75.26 ± 0.79 ^A^	1584.1

^1^ Average value ± SD, n = 3. ^2^ The energy content was calculated on Atwater factors. Values in a column with the same letter are not statistically different at a 5% significance level, according to Duncan’s test.

**Table 3 plants-13-00143-t003:** Total polyphenols, flavonoids, carotenoids contents, and antioxidant activity of maize hybrids across studied FAO maturity groups.

Maturity Group	Sample Name	Total Polyphenols (mg GA eq kg^−1^) ^1^	Flavonoids (mg Q eq kg^−1^) ^1^	DPPH (mg AA eq g^−1^ DM) ^1^	Carotenoids (mg β-C eq kg^−1^) ^1^
	G11	1377.12 ± 7.57 ^GH^	1090.76 ± 12.93 ^J^	0.20 ± 0.006 ^GH^	53.80 ± 1.03 ^L^
	G24	1136.34 ± 7.39 ^L^	1023.82± 3.82 ^MN^	0.13 ± 0.006 ^M^	71.68 ± 0.90 ^G^
FAO 100–200	G27	1067.52 ± 19.59 ^N^	981.54 ± 11.86 ^O^	0.14 ± 0.006 ^LM^	83.93 ± 2.12 ^C^
	Minimal value	1067.52 ± 19.59 ^N^	981.54 ± 11.86 ^O^	0.13 ± 0.006 ^M^	53.80 ± 1.03 ^L^
	Maximal value	1377.12 ± 7.57 ^GH^	1090.76 ± 12.93 ^J^	0.20 ± 0.006 ^GH^	83.93 ± 2.12 ^C^
	G4	1560.22 ± 14.92 ^C^	1550.40 ± 3.92 ^B^	0.29± 0.018 ^B^	80.48 ± 1.62 ^D^
	G6	1485.31 ± 12.12 ^E^	1094.59 ± 14.85 ^IJ^	0.27 ± 0.006 ^C^	64.79 ± 1.68 ^HI^
	G8	1437.35 ± 18.75 ^F^	1172.41 ± 19.67 ^F^	0.23 ± 0.011 ^DE^	65.27 ± 0.92 ^HI^
	G10	1392.44 ± 7.46 ^G^	973.18 ± 9.41 ^O^	0.20 ± 0.006 ^FGH^	58.34 ± 1.054 ^K^
FAO 200–400	G15	1314.13 ± 9.13 ^I^	1073.42 ± 7.99 ^JK^	0.23 ± 0.006 ^DE^	75.29 ± 0.70 ^F^
	G19	1240.27 ± 9.32 ^J^	1000.28 ± 9.84 ^NO^	0.16 ± 0.006 ^JK^	66.28 ± 0.77 ^H^
	G26	1092.78 ± 16.41 ^M^	787.66 ± 9.79 ^R^	0.14 ± 0.006 ^LM^	75.92 ± 1.62 ^EF^
	Minimal value	1092.78 ± 16.41 ^M^	787.66 ± 9.79 ^R^	0.14 ± 0.006 ^LM^	58.34 ± 1.054 ^K^
	Maximal value	1560.22 ± 14.92 ^C^	1550.40 ± 3.92 ^B^	0.29± 0.018 ^B^	80.48 ± 1.62 ^D^
	G2	1626.70 ± 6.06 ^B^	1226.20 ± 6.40 ^E^	0.08 ± 0.012 ^OP^	74.70 ± 1.91 ^F^
	G5	1519.04 ± 12.30 ^D^	1482.53 ± 2.01 ^C^	0.27 ± 0.006 ^BC^	60.67 ± 1.11 ^JK^
	G7	1477.33 ± 6.48 ^E^	1293.45 ± 12.76 ^D^	0.28 ± 0.006 ^BC^	61.51 ± 0.73 ^J^
	G9	1418.16 ± 6.03 ^F^	1142.33 ± 17.80 ^GH^	0.24 ± 0.006 ^D^	58.29 ± 1.97 ^K^
	G17	1263.80 ± 22.16 ^J^	1166.42 ± 9.69 ^FG^	0.18 ± 0.012 ^GHI^	75.92 ± 0.17 ^EF^
FAO 400–600	G20	1238.73 ± 13.30 ^J^	1118.42 ± 19.29 ^HI^	0.15 ± 0.006 ^KL^	63.79 ± 0.70 ^HIJ^
	G22	1160.37 ± 5.91 ^KL^	862.14 ± 13.59 ^Q^	0.18 ± 0.012 ^IJ^	98.15 ± 2.94 ^A^
	G28	1043.76 ± 10.09 ^N^	866.83 ± 22.70 ^Q^	0.10 ± 0.006 ^NO^	86.18 ± 0.68 ^C^
	G30	711.86 ± 2.98 ^P^	686.61 ± 7.83 ^S^	0.07 ± 0.006 ^P^	70.73 ± 2.13 ^G^
	Minimal value	711.86 ± 2.98 ^P^	686.61 ± 7.83 ^S^	0.07 ± 0.006 ^P^	58.29 ± 1.97 ^K^
	Maximal value	1626.70 ± 6.06 ^B^	1482.53 ± 2.01 ^C^	0.28 ± 0.006 ^BC^	98.15 ± 2.94 ^A^
	G1	1720.14 ± 20.48 ^A^	1159.99 ± 18.47 ^FG^	0.38 ± 0.006 ^A^	65.35 ± 0.06 ^HI^
	G3	1607.05 ± 15.79 ^B^	1577.59 ± 16.94 ^A^	0.25 ± 0.006 ^D^	78.80 ± 2.51 ^DE^
	G12	1360.79 ± 7.46 ^H^	1043.75 ± 11.75 ^LM^	0.21 ± 0.006 ^EF^	65.64 ± 1.73 ^HI^
	G13	1354.53 ± 13.10 ^H^	1120.92 ± 11.24 ^HI^	0.19 ± 0.011 ^GHI^	16.99 ± 1.38 ^N^
	G14	1353.40 ± 8.95 ^H^	1156.28 ± 9.28 ^FG^	0.19 ± 0.012 ^GHI^	43.49 ± 0.42 ^M^
	G16	1312.91 ± 11.82 ^I^	1140.42 ± 3.53 ^GH^	0.18 ± 0.012 ^IJ^	12.85 ± 1.59 ^O^
FAO 600–700	G18	1248.13 ± 8.86 ^J^	1240.44 ± 5.82 ^E^	0.22 ± 0.012 ^EF^	66.15 ± 1.22 ^H^
	G21	1178.36 ± 4.52 ^K^	1138.13 ± 11.86 ^GH^	0.17 ± 0.006 ^IJ^	91.27 ± 0.95 ^B^
	G23	1140.91 ± 2.97 ^L^	698.36 ± 17.53 ^S^	0.18 ± 0.006 ^HIJ^	64.89 ± 0.31 ^HI^
	G25	1134.25 ± 10.34 ^L^	1062.62 ± 14.99 ^KL^	0.13 ± 0.012 ^M^	63.18 ± 1.27 ^HIJ^
	G29	938.72 ± 16.41 ^O^	898.40 ± 13.70 ^P^	0.11 ± 0.012 ^N^	62.59 ± 1.16 ^IJ^
	Minimal value	938.72 ± 16.41 ^O^	698.36 ± 17.53 ^S^	0.11 ± 0.012 ^N^	12.85 ± 1.59 ^O^
	Maximal value	1720.14 ± 20.48 ^A^	1577.59 ± 16.94 ^A^	0.38 ± 0.006 ^A^	91.27 ± 0.95 ^B^

^1^ Average value ± SD, n = 3. Values with the same letter in a column are not significantly different at 5%, according to Duncan’s test.

**Table 4 plants-13-00143-t004:** Tocopherol content of maize hybrids across the studied FAO groups.

Maturity Group	Sample Name	α-T (mg kg^−1^) ^1^	β-T (mg kg^−1^) ^1^	γ-T (mg kg^−1^) ^1^	δ-T (mg kg^−1^) ^1^	Total (mg kg^−1^) ^1^
	G11	12.38 ± 0.02 ^MN^	0.00 ± 0.00 ^J^	62.22 ± 0.49 ^U^	0.00 ± 0.00 ^M^	75.06 ± 0.14 ^W^
	G24	5.17 ± 0.08 ^Q^	0.00 ± 0.00 ^J^	127.87 ± 1.03 ^M^	8.76 ± 0.16 ^G^	142.92 ± 0.31 ^O^
FAO 100–200	G27	31.09 ± 0.18 ^H^	0.00 ± 0.00 ^J^	255.65 ± 0.73 ^G^	10.13 ± 0.34 ^G^	295.73 ± 2.50 ^I^
	Minimal value	5.17 ± 0.08 ^Q^	0.00 ± 0.00 ^J^	62.22 ± 0.49 ^U^	0.00 ± 0.00 ^M^	75.06 ± 0.14 ^W^
	Maximal value	31.09 ± 0.18 ^H^	0.00 ± 0.00 ^J^	255.65 ± 0.73 ^G^	10.13 ± 0.34 ^G^	295.73 ± 2.50 ^I^
	G4	0.00 ± 0.00 ^T^	13.85 ± 0.39 ^D^	5.35 ± 0.45 ^a^	0.00 ± 0.00 ^M^	19.60 ± 0.27 ^d^
	G6	31.77 ± 0.40 ^H^	0.00 ± 0.00 ^J^	423.13 ± 0.93 ^B^	17.50 ± 0.54 ^D^	472.65 ± 0.72 ^C^
	G8	120.17 ± 0.99 ^C^	0.00 ± 0.00 ^J^	351.86 ± 1.05 ^C^	26.75 ± 0.40 ^B^	500.03 ± 0.69 ^B^
	G10	13.57 ± 0.99 ^L^	0.00 ± 0.00 ^J^	129.15 ± 0.52 ^L^	7.60 ± 0.27 ^I^	150.96 ± 0.52 ^M^
FAO 200–400	G15	9.00 ± 0.15 ^O^	14.34 ± 0.31 ^C^	12.65 ± 0.16 ^Z^	0.00 ± 0.00 ^M^	36.65 ± 0.31 ^b^
	G19	6.89 ± 0.29 ^P^	0.00 ± 0.00 ^J^	73.99 ± 0.29 ^T^	8.99 ± 0.17 ^G^	89.97 ± 0.61 ^U^
	G26	3.80 ± 0.08 ^R^	9.52 ± 0.40 ^F^	83.62 ± 0.58 ^S^	6.65 ± 0.70 ^J^	103.95 ± 0.11 ^R^
	Minimal value	0.00 ± 0.00 ^T^	0.00 ± 0.00 ^J^	5.35 ± 0.45 ^a^	0.00 ± 0.00 ^M^	19.60 ± 0.27 ^d^
	Maximal value	120.17 ± 0.99 ^C^	14.34 ± 0.31 ^C^	423.13 ± 0.93 ^B^	26.75 ± 0.40 ^B^	500.03 ± 0.69 ^B^
	G2	13.09 ± 0.52 ^LM^	0.00 ± 0.00 ^J^	55.11 ± 0.33 ^W^	0.00 ± 0.00 ^M^	67.97 ± 0.13 ^X^
	G5	36.04 ± 0.51 ^G^	0.00 ± 0.00 ^J^	113.73 ± 0.86 ^Q^	0.00 ± 0.00 ^M^	150.43 ± 0.44 ^M^
	G7	135.11 ± 0.30 ^B^	28.96 ± 0.31 ^A^	134.50 ± 0.31 ^K^	30.28 ± 0.22 ^A^	328.46 ± 0.47 ^H^
	G9	4.05 ± 0.23 ^R^	0.00 ± 0.00 ^J^	118.90 ± 0.73 ^P^	5.28 ± 0.23 ^K^	127.84 ± 0.29 ^P^
	G17	7.00 ± 0.17 ^P^	0.00 ± 0.00 ^J^	53.69 ± 0.45 ^X^	5.39 ± 0.38 ^K^	66.07 ± 0.21 ^Y^
FAO 400–600	G20	18.11 ± 0.14 ^J^	6.58 ± 0.41 ^I^	195.10 ± 0.33 ^I^	19.60 ± 0.67 ^C^	239.22 ± 0.68 ^J^
	G22	45.37 ± 0.48 ^F^	0.00 ± 0.00 ^J^	284.12 ± 0.19 ^F^	6.31 ± 0.27 ^J^	336.84 ± 0.54 ^G^
	G28	15.95 ± 0.09 ^K^	0.00 ± 0.00 ^J^	83.92 ± 0.29 ^S^	0.00 ± 0.00 ^M^	99.87 ± 0.20 ^S^
	G30	12.01 ± 0.05 ^N^	0.00 ± 0.00 ^J^	84.19 ± 0.43 ^S^	0.00 ± 0.00 ^M^	96.83 ± 0.41 ^T^
	Minimal value	4.05 ± 0.23 ^R^	0.00 ± 0.00 ^J^	53.69 ± 0.45 ^X^	0.00 ± 0.00 ^M^	66.07 ± 0.21 ^Y^
	Maximal value	135.11 ± 0.30 ^B^	28.96 ± 0.31 ^A^	284.12 ± 0.19 ^F^	30.28 ± 0.22 ^A^	336.84 ± 0.54 ^G^
	G1	205.42 ± 0.77 ^A^	0.00 ± 0.00 ^J^	478.71 ± 0.56 ^A^	16.02 ± 0.19 ^E^	701.11 ± 0.18 ^A^
	G3	134.82 ± 0.60 ^B^	0.00 ± 0.00 ^J^	196.36 ± 0.21 ^H^	10.29 ± 0.22 ^G^	341.11 ± 0.34 ^F^
	G12	3.28 ± 0.10 ^RS^	18.58 ± 0.57 ^B^	5.15 ± 0.26 ^a^	0.00 ± 0.00 ^M^	26.44 ± 0.59 ^c^
	G13	36.96 ± 1.19 ^G^	7.51 ± 0.39 ^H^	181.97 ± 0.70 ^J^	7.55 ± 0.43 ^I^	235.19 ± 1.02 ^K^
	G14	64.10 ± 0.45 ^E^	0.00 ± 0.00 ^J^	294.37 ± 0.54 ^E^	0.00 ± 0.00 ^M^	358.64 ± 0.74 ^E^
	G16	7.87 ± 0.08 ^P^	0.00 ± 0.00 ^J^	103.78 ± 0.91 ^R^	4.32 ± 0.10 ^L^	116.99 ± 0.35 ^Q^
FAO 600–700	G18	2.71 ± 0.38 ^RS^	0.00 ± 0.00 ^J^	48.19 ± 0.10 ^Y^	0.00 ± 0.00 ^M^	50.37 ± 0.47 ^a^
	G21	0.00 ± 0.00 ^T^	0.00 ± 0.00 ^J^	47.98 ± 0.30 ^Y^	5.28 ± 0.10 ^K^	54.20 ± 0.92 ^Z^
	G22	45.37 ± 0.48 ^F^	0.00 ± 0.00 ^J^	284.12 ± 0.19 ^F^	6.31 ± 0.27 ^J^	336.84 ± 0.54 ^G^
	G23	28.44 ± 0.45 ^I^	0.00 ± 0.00 ^J^	126.27 ± 0.37 ^N^	7.25 ± 0.05 ^I^	162.77 ± 0.27 ^L^
	G25	106.27 ± 0.79 ^D^	11.90 ± 0.72 ^E^	316.10 ± 0.89 ^D^	13.54 ± 0.47 ^F^	447.85 ± 0.41 ^D^
	G29	13.36 ± 0.54 ^LM^	8.99 ± 0.17 ^G^	124.57 ± 0.35 ^O^	0.00 ± 0.00 ^M^	147.34 ± 0.46 ^N^
	Minimal value	0.00 ± 0.00 ^T^	0.00 ± 0.00 ^J^	5.15 ± 0.26 ^a^	0.00 ± 0.00 ^M^	26.44 ± 0.59 ^c^
	Maximal value	205.42 ± 0.77 ^A^	18.58 ± 0.57 ^B^	478.71 ± 0.56 ^A^	16.02 ± 0.19 ^E^	701.11 ± 0.18 ^A^

^1^ Average value ± SD, n = 2. Values in a column with the same letter are not statistically different at a 5% significance level according to Duncan’s test.

**Table 5 plants-13-00143-t005:** Composition of sought-after seed oil fatty acids (% composition by weight) across the studied FAO groups.

Fatty Acid		Fatty Acid Content (%)		
	FAO 100–200 ^1^	FAO 200–400 ^1^	FAO 400–600 ^1^	FAO 600–700 ^1^
C14:0	0.01 ± 0.01 ^A^	0.02 ± 0.02 ^A^	0.01 ± 0.01 ^A^	0.01 ± 0.01 ^A^
C16:0	11.81 ± 0.63 ^A^	11.43 ± 1.11 ^AB^	10.84 ± 0.81 ^B^	11.42 ± 1.04 ^AB^
C18:0	0.60 ± 0.16 ^A^	0.60 ± 0.10 ^A^	0.58 ± 0.09 ^A^	0.56 ± 0.09 ^A^
C18:1	34.78 ± 4.85 ^A^	34.19 ± 3.15 ^A^	33.29 ± 2.32 ^A^	32.51 ± 2.22 ^A^
C18:2	52.17 ± 4.43 ^A^	51.42 ± 3.39 ^A^	50.28 ± 2.48 ^A^	49.68 ± 2.41 ^A^
C18:3	1.50 ± 0.47 ^B^	1.88 ± 0.69 ^AB^	2.11 ± 0.65 ^A^	1.70 ± 0.36 ^AB^
C20:0	0.44 ± 0.11 ^A^	0.37 ± 0.11 ^A^	0.37 ± 0.08 ^A^	0.43 ± 0.17 ^A^
C20:1	0.70 ± 0.13 ^A^	0.73 ± 0.17 ^A^	0.77 ± 0.16 ^A^	0.64 ± 0.17 ^A^
C22:0	0.25 ± 0.03 ^B^	0.30 ± 0.09 ^AB^	0.36 ± 0.12 ^A^	0.29 ± 0.09 ^AB^
C24:0	0.22 ± 0.17 ^A^	0.21 ± 0.09 ^A^	0.25 ± 0.08 ^A^	0.24 ± 0.07 ^A^
SFA	13.33 ± 0.80 ^A^	12.92 ± 1.10 ^AB^	12.41 ± 0.82 ^B^	12.95 ± 1.10 ^AB^
UFA	86.67 ± 0.80 ^B^	87.08 ± 1.10 ^AB^	87.59 ± 0.82 ^A^	87.02 ± 1.10 ^AB^
MUFA	35.48 ± 4.88 ^A^	34.92 ± 3.24 ^A^	34.06 ± 2.30 ^A^	33.15 ± 2.28 ^A^
PUFA	51.19 ± 4.45 ^A^	52.16 ± 3.25 ^AB^	53.53 ± 2.51 ^AB^	53.87 ± 2.28 ^A^

^1^ Average value ± SD, n = 2. Values in a column with the same letter are not statistically different at a 5% significance level, according to Duncan’s test. SFA—saturated fatty acids, UFA—unsaturated fatty acids, MUFA—monounsaturated fatty acids, PUFA—polyunsaturated fatty acids.

## Data Availability

Data will be made available on demand.

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
