# Peer review of "Nutritional and Chemical Quality of Maize Hybrids from Different FAO Maturity Groups Developed and Grown in Serbia"

_plants, 2024, doi:10.3390/plants13010143_

Round 1
Reviewer 1 Report
Comments and Suggestions for Authors
The paper entitled "Nutritional and Chemical Quality of Maize Hybrids from Different FAO Maturity Groups Developed and Grown in Serbia" presents a comprehensive analysis of maize hybrids from different FAO maturity groups in Serbia, exploring variations in polyphenols, flavonoids, carotenoids, tocopherols and fatty acids.
One of the notable strengths of this paper is its clear and structured organization. The introduction briefly outlines the research problem and its significance, providing the reader with a solid basis for understanding the context. The methodology section is detailed and methodically presented, allowing for the replication of experiments and ensuring the credibility of the findings.
The authors have meticulously analyzed the data collected, utilizing appropriate statistical methods and robust analytical tools. The results are presented logically. Furthermore, the discussion section effectively interprets the results.
The paper's language is precise, and the writing style is understandable. The use of figures and tables enhances the clarity of the information presented.
The well-structured approach, rigorous methodology and clear presentation of results make this work a valuable addition to the existing literature in this field. Therefore, I strongly encourage the publication of this paper in present form.
Author Response
Replies for
Referee 1
Authors comment
We express gratitude for your dedicated effort and the time invested in reviewing the manuscript. Furthermore, we are delighted that you found favor in both the organization of our manuscript and the presentation of the obtained results.
Sincerely yours,
PhD Nada Grahovac, Senior Research Associate
Corresponding author: Dragan Živančev
Institute of Field and Vegetable Crops,
Maksima Gorkog 30, 21101 Novi Sad, Serbia
Tel: +381 21 4898 321; Fax: +381 21 4898 418
E-mail address: [email protected]
Resubmission Date
21 December 2023
Reviewer 2 Report
Comments and Suggestions for Authors
The research work "Nutritional and chemical quality of maize hybrids from different FAO maturity groups developed and grown in Serbia" is quite interesting. The introduction to the scientific work was presented in a correct and understandable way. However, there is no separation of materials and methods. They are briefly described in the remaining chapters. However, it is much easier if the chapter is presented separately. The results and discussion chapter is presented correctly. Only the materials and methods section should, in my opinion, be earlier. This would make reading research work easier. The tables were presented correctly and legibly. The added value to the article are the figures. Interesting article. The whole thing is full of results. The result of annual research. It is worth checking the following years. What agricultural technology was used? Recommended for publication, after minor corrections.
Best regards
Magdalena Sobolewska

Author Response
Replies for
Referee 2
Authors comments
Thank you very much for your consideration and invaluable comments on our recent revision. Your suggestions and feedback on the content helped us to improve the manuscript. We appreciate your effort and the time you took to read the manuscript. We carefully went through all of your comments and revised the manuscript accordingly. We have revised, taking all your comments into account, added content and re-organized the rewriting.
Referee comments and Text Location |
Comment |
Line 1-155 in Materials and Methods “However, there is no separation of materials and methods. They are briefly described in the remaining chapters. However, it is much easier if the chapter is presented separately.” |
We have adopted the referee's recommendation to relocate the materials and methods section ahead of the results and discussion. In the revised manuscript, you will find the materials and methods now positioned after the introductory section. |
What agricultural technology was used? |
Following the referee's request, we have incorporated a section detailing the agricultural technology utilized. “Winter wheat (Triticum aestivum L.) was the previous crop. The selected plots underwent plowing in October to a depth of up to 30 cm, followed by seedbed preparation before sowing in March using heavy-duty cultivators (Multi-Tiller) at a depth of 15 cm. On 22 April 2021, the crop was sown using a Wintersteiger AG pneumatic precision seed drill at a depth of 5 cm. The plot dimensions were 5 × 2.8 m, featuring intra-row spacing of 22 cm and row spacing of 70 cm. Throughout both years, weed control was executed through conventional chemical methods. Pre-emergence application included a dose of 1.4 l ha-1 combined with 3.5 L ha-1 a mixture containing 375 g L-1 S-Metolachlor, 125 g l-1 Terbuthylazine and 37.5 g l-1 Mesotrione. Post-emergence application was also conducted. During the vegetation season, control of Sorghum halepense sp. and other narrow-leaved weeds was achieved by applying Nicosulfuron or Rimsulfuron at a rate of 50-60 g ha-1.” |
Sincerely yours,
PhD Nada Grahovac, Senior Research Associate
Corresponding author: Nada Grahovac
Institute of Field and Vegetable Crops,
Maksima Gorkog 30, 21101 Novi Sad, Serbia
Tel: +381 21 4898 321; Fax: +381 21 4898 418
E-mail address: [email protected]
Resubmission Date
21 December 2023

Reviewer 3 Report
Comments and Suggestions for Authors
The work presented here enables us to compare the chemical composition of certain hydrides of maize and to highlight their health benefits. The chemical analyses appear to have been carried out with precision and repeatability, given the procedures described. However, the purpose or interest of studying these varieties is not defined. What was the purpose of these studies? Certainly not to "provide information on the impact of agro-ecological conditions on the nutritional potential of maize hybrids", as is stated; because the results do not allow us to differentiate between agro-ecological conditions: How do these varieties reduce erosion and tillage, increase soil fertility, increase functional biodiversity, preserve water resources, adapt to climate change....
Furthermore, why study these varieties: for their future use (grain, forage, waxy, oil-rich, …), their earliness, their yield? Do they perform well enough compared to the range of most widely used commercial varieties?
Are these studies carried out with certification in mind?
Introduction
· Lines 49-52: The upper oil content of maize is first proposed at 4%, then it said that a content of 5,50% until 9,71% could be reached. These sentences are not clear and references were missing.
· Line 52-54: precise what are the factors which justify the FA percentages: variety, agroclimatic conditions…in regard of the cited reference.
· Rewrite the paragraph from line 49 to line 76 to correctly describe the chemical composition of maize, dedicating a paragraph to each family of compounds (fatty acids, tocopherols, polyphenols, etc.), giving their benefits for human health, the range of variations in their content and the factors that can influence them, particularly in the case of hybrid maize since this is the type of sample studied in this work.
· At the end of the introduction, it said that “The results obtained from this study provide insights into how agro-ecological conditions impact the nutritional potential of maize hybrids.”: but what were the agro-ecological conditions studied here? Results showed only differences in the chemical composition according to hybrid maize samples growing under the same agricultural practices.
Results:
· What do the FAO maturity groups correspond to? Can you give a brief description and cite references.
· Table 2: remind FAO maturity group for studied hybrid maize sample or group them according FAO maturity group for memory. What is the range, the dispersion of chemical data in the same FAO maturity group? Give the minimum and the maximum values.
· Same remarks for table 3.
· Why FA contents were not presented for each hybrid maize samples; only an average value of FA contents were presented according to FAO maturity groups…why? All maize oils obtained from hybrid samples of a same group have been mixed???
· Lines 204-205: Cite reference about the benefits of the ratio of n-6 and n-3 acids.
· Lines 216-243: in my opinion, this part is unnecessary if a PCA has been carried out all chemical data. Figure 1 is sufficient in itself to serve as a support for determining correlated or anti-correlated variables. Add a color code according to the FAO maturity group. Revised the legend of variable by removing units of chemical data (for example eq quertecin, etc) and give a detailed legend of used acronyms. Why SFA, UFA, MUFA and PUFA were not considered as variables in PCA? Why use crude fat ?
Material and methods
· Sample preparation: Were the maize samples dried before grinding?
· What were the soil cultivation practices applied? What were the climatic conditions during the growing of the maize samples?
· How many days after sowing, was the maize harvested?
Author Response
Replies for
Referee 3
Authors comments
Thank you very much for your consideration and invaluable comments on our recent revision. Your suggestions and feedback on the content helped us to improve the manuscript. We appreciate your effort and the time you took to read the manuscript. We carefully went through all of your comments and revised the manuscript accordingly. We have revised, taking all your comments into account, added content and re-organized the rewriting.
Referee comments and Text Location |
Comment |
However, the purpose or interest of studying these varieties is not defined. What was the purpose of these studies? Certainly not to "provide information on the impact of agro-ecological conditions on the nutritional potential of maize hybrids", as is stated; because the results do not allow us to differentiate between agro-ecological conditions: How do these varieties reduce erosion and tillage, increase soil fertility, increase functional biodiversity, preserve water resources, adapt to climate change.” |
Following the referee's request, we have inserted purpose of studying these varieties. “The purpose of this research is to investigate and understand the proximate analysis of maize hybrids from different FAO maturity groups in Serbia. The results of the study reveal substantial variations in the nutritional composition and antioxidant properties among maize hybrids from different FAO maturity groups. The levels of total polyphenols, flavonoids, carotenoids, tocopherols, and fatty acids varied significantly, indicating that specific hybrids within different maturity groups may offer distinct health benefits. The emphasis on maturity groups underscores the importance of considering this factor when assessing the nutritional potential of maize hybrids. Overall, the research provides valuable insights into the diversity of nutritional content in maize hybrids, with implications for both food production and industrial applications” (in introduction section). |
Furthermore, why study these varieties: for their future use (grain, forage, waxy, oil-rich, …), their earliness, their yield? Do they perform well enough compared to the range of most widely used commercial varieties? |
We agreed with referee’s observation and we have inserted in material and methods. “Maize hybrids of standard grain quality with different vegetation lengths and production purposes, as well as hybrids with specific properties.” (First paragraph 2.1. Meterial) |
Are these studies carried out with certification in mind? |
These are commercial hybrids. |
Lines 49-52: The upper oil content of maize is first proposed at 4%, then it said that a content of 5,50% until 9,71% could be reached. These sentences are not clear and references were missing. |
We agreed with referee’s opinion that we have rewrited paragraph 49-52 in introduction section, second paragraph. “Maize, from a nutritional standpoint, boasts a noteworthy chemical composition that distinguishes it among cereals. Typically, the oil content in maize kernels ranges from 2% to 6%, with approximately 85% primarily concentrated in the germ's scutel-lum [6]. While oil constitutes a minor fraction of the grain, its quantity can differ among maize varieties with distinct genetic backgrounds [7]. The lipids in maize can be categorized as nonpolar, polar, and non-saponifiable [8]. The most prevalent type is the nonpolar fraction, which includes fatty acids (FA) and triglycerides rich in linoleic acid (18:2), followed by oleic acid (18:1) (OFA). Linoleic acid (LFA) is recognized as the singular essential fatty acid for humans, and maize oil comprises over 50% of this spe-cific FA. About 99% of maize oil composition comprises fatty acids such as palmitic (16:0), stearic (18:0), oleic (18:1), linoleic (18:2), and linolenic (18:3) acids [9]. Among these, monounsaturated oleic acid (OLA) constitutes 24%, and polyunsaturated linole-ic acid (LOA) constitutes 62% [6]. Consuming crude or refined corn oils rich in linoleic acid (LOA) enhances the balance between saturated and unsaturated dietary fats. Re-grettably, it does not impact the ratio between omega-6 and omega-3 fatty acids (FA). Linoleic acid (18:3) can undergo conversion in the human system to linolenic acid (LLA) and subsequently to other significant long-chain fatty acids and metabolites, in-cluding eicosanoids and prostaglandins [6]. From a nutraceutical perspective, polyun-saturated omega-3 fatty acids are acknowledged for their ability to decrease serum cholesterol. The percentages of unsaturated fatty acids (UFA), particularly oleic and linoleic acids, in maize oil are influenced by various factors, including maize variety, agroclimatic conditions, geographical location, agronomic factors and harvesting and processing methods. Respectively, the percentages of unsaturated fatty acids in maize oil are a result of complex interactions between the genetic makeup of the maize vari-ety, the environmental conditions in which it is grown, and various agricultural and processing practices. The protein content in maize, a crucial nutritional component, usually hovers around 10%, providing a valuable source of dietary protein. Corn oil is acknowledged as an exceptional source of tocopherols, which serve as antioxidants and contribute significantly to vitamin E intake. Similar to essential fatty acids, vitamin E is a vital component of the human diet as the body cannot synthesize it internally. Tocopherols, a robust class of antioxidants, play a crucial role in safeguarding polyunsaturated fatty acids within membranes from degradation caused by reactive oxygen species like ozone, singlet oxygen, peroxides, and hyperoxides [10]. The antioxidant function of tocopherols is not only pivotal for health but also essential for maintaining oil quality, extending its shelf life by inhibiting the onset of rancidity. Corn oil typically contains four major tocopherols: alpha-, beta-, gamma-, and delta-tocopherol. Among these, gamma-tocopherol predominates in commercially available corn oil, followed by al-pha-tocopherol and delta-tocopherol. Delta-tocopherol, in particular, exhibits the most potent antioxidant effect, while alpha-tocopherol boasts the highest vitamin E activity. Maize displays a diverse range of colors, spanning from white to yellow, red, blue, purple, and more. Kernels with blue, purple, and red hues are particularly abundant in anthocyanins, known for their well-established antioxidant and bioactive properties [11]. The content of anthocyanin, carotenoid, and phenolic compounds in maize varies according to its coloration. The pericarp portion holds the highest concentration of anthocyanin pigments, while the aleurone layer contains smaller amounts [12]. Maize exhibits notable natural variability in kernel carotenoids, with certain genotypes ac-cumulating as much as 66.0 μg g-1 [13]. Floury maize tends to have fewer carotenoids compared to yellow maize. Typically, provitamin A carotenoids make up only 10–20% of the total carotenoid content in maize, while zeaxanthin and lutein each represent 30–50%. In standard maize varieties, concentrations of provitamin A carotenoids, namely α-carotene, β-carotene, and β-cryptoxanthin, range from 0 to 1.3, 0.13 to 2.7, and 0.13 to 1.9 nmol g-1, respectively [14]. In its natural state, maize serves as an abun-dant reservoir of carotenoids, including beta-carotene, zeaxanthin, lutein, and crypto-xanthin. These carotenoids offer a wide array of health benefits, spanning from the preservation of normal vision to the reduction of oxidative stress. Maize is recognized for its rich content of various phenolic acids, with ferulic acid standing out as a crucial phytochemical whose concentration varies among different maize types. High-carotenoid maize, in particular, exhibits a higher total ferulic acid content compared to white, yellow, red, and blue maize varieties. It's noteworthy that a significant portion of the ferulic acid in maize exists in a bound form [11]. The majority of phenolics, including phenolic acids, flavonoids, and conjugated amines, are pre-dominantly concentrated in the pericarp and aleurone layers, as well as the germ, with minimal traces found in the endosperm [15]. Total polyphenols, found in maize ker-nels, possess antidiabetic, antiobesity, and anticancer effects, contributing to the preservation of the human cardiovascular system. The chemical composition of maize, especially in hybrid varieties studied in this research, is subject to variations influenced by genetic factors, agroclimatic conditions, and specific breeding practices. Understanding these factors is crucial for elucidating the nutritional potential and health benefits associated with consuming different maize hybrids.“ |
Line 52-54: precise what are the factors which justify the FA percentages: variety, agroclimatic conditions…in regard of the cited reference. |
According referee’s opinion we added factors which justify the FA percentages in introduction section, second paragraph. “The percentages of unsaturated fatty acids (UFA), particularly oleic and linoleic acids, in maize oil are influenced by various factors, including maize variety, agroclimatic conditions, geographical location, agronomic factors and harvesting and processing methods. Respectively, the percentages of unsaturated fatty acids in maize oil are a result of complex interactions between the genetic makeup of the maize variety, the environmental conditions in which it is grown, and various agricultural and processing practices. ” in introduction section. |
Rewrite the paragraph from line 49 to line 76 to correctly describe the chemical composition of maize, dedicating a paragraph to each family of compounds (fatty acids, tocopherols, polyphenols, etc.), giving their benefits for human health, the range of variations in their content and the factors that can influence them, particularly in the case of hybrid maize since this is the type of sample studied in this work. |
According referee’s observation we rewrited the paragraph from line 49 to line 76 in introduction section. “Maize, from a nutritional standpoint, boasts a noteworthy chemical composition that distinguishes it among cereals. Typically, the oil content in maize kernels ranges from 2% to 6%, with approximately 85% primarily concentrated in the germ's scutellum [6]. While oil constitutes a minor fraction of the grain, its quantity can differ among maize varieties with distinct genetic backgrounds [7]. The lipids in maize can be categorized as nonpolar, polar, and non-saponifiable [8]. The most prevalent type is the nonpolar fraction, which includes fatty acids (FA) and triglycerides rich in linoleic acid (18:2), followed by oleic acid (18:1) (OFA). Linoleic acid (LFA) is recognized as the singular essential fatty acid for humans, and maize oil comprises over 50% of this specific FA. About 99% of maize oil composition comprises fatty acids such as palmitic (16:0), stearic (18:0), oleic (18:1), linoleic (18:2), and linolenic (18:3) acids [9]. Among these, monounsaturated oleic acid (OLA) constitutes 24%, and polyunsaturated linoleic acid (LOA) constitutes 62% [6]. Consuming crude or refined corn oils rich in linoleic acid (LOA) enhances the balance between saturated and unsaturated dietary fats. Regrettably, it does not impact the ratio between omega-6 and omega-3 fatty acids (FA). Linoleic acid (18:3) can undergo conversion in the human system to linolenic acid (LLA) and subsequently to other significant long-chain fatty acids and metabolites, including eicosanoids and prostaglandins [6]. From a nutraceutical perspective, polyunsaturated omega-3 fatty acids are acknowledged for their ability to decrease serum cholesterol. The percentages of unsaturated fatty acids (UFA), particularly oleic and linoleic acids, in maize oil are influenced by various factors, including maize variety, agroclimatic conditions, geographical location, agronomic factors and harvesting and processing methods. Respectively, the percentages of unsaturated fatty acids in maize oil are a result of complex interactions between the genetic makeup of the maize variety, the environmental conditions in which it is grown, and various agricultural and processing practices. The protein content in maize, a crucial nutritional component, usually hovers around 10%, providing a valuable source of dietary protein. Corn oil is acknowledged as an exceptional source of tocopherols, which serve as antioxidants and contribute significantly to vitamin E intake. Similar to essential fatty acids, vitamin E is a vital component of the human diet as the body cannot synthesize it internally. Tocopherols, a robust class of antioxidants, play a crucial role in safeguarding polyunsaturated fatty acids within membranes from degradation caused by reactive oxygen species like ozone, singlet oxygen, peroxides, and hyperoxides [10]. The antioxidant function of tocopherols is not only pivotal for health but also essential for maintaining oil quality, extending its shelf life by inhibiting the onset of rancidity. Corn oil typically contains four major tocopherols: alpha-, beta-, gamma-, and delta-tocopherol. Among these, gamma-tocopherol predominates in commercially available corn oil, followed by alpha-tocopherol and delta-tocopherol. Delta-tocopherol, in particular, exhibits the most potent antioxidant effect, while alpha-tocopherol boasts the highest vitamin E activity. Maize displays a diverse range of colors, spanning from white to yellow, red, blue, purple, and more. Kernels with blue, purple, and red hues are particularly abundant in anthocyanins, known for their well-established antioxidant and bioactive properties [11]. The content of anthocyanin, carotenoid, and phenolic compounds in maize varies according to its coloration. The pericarp portion holds the highest concentration of anthocyanin pigments, while the aleurone layer contains smaller amounts [12]. Maize exhibits notable natural variability in kernel carotenoids, with certain genotypes accumulating as much as 66.0 μg g-1 [13]. Floury maize tends to have fewer carotenoids compared to yellow maize. Typically, provitamin A carotenoids make up only 10–20% of the total carotenoid content in maize, while zeaxanthin and lutein each represent 30–50%. In standard maize varieties, concentrations of provitamin A carotenoids, namely α-carotene, β-carotene, and β-cryptoxanthin, range from 0 to 1.3, 0.13 to 2.7, and 0.13 to 1.9 nmol g-1, respectively [14]. In its natural state, maize serves as an abundant reservoir of carotenoids, including beta-carotene, zeaxanthin, lutein, and cryptoxanthin. These carotenoids offer a wide array of health benefits, spanning from the preservation of normal vision to the reduction of oxidative stress. Maize is recognized for its rich content of various phenolic acids, with ferulic acid standing out as a crucial phytochemical whose concentration varies among different maize types. High-carotenoid maize, in particular, exhibits a higher total ferulic acid content compared to white, yellow, red, and blue maize varieties. It's noteworthy that a significant portion of the ferulic acid in maize exists in a bound form [11]. The majority of phenolics, including phenolic acids, flavonoids, and conjugated amines, are predominantly concentrated in the pericarp and aleurone layers, as well as the germ, with minimal traces found in the endosperm [15]. Total polyphenols, found in maize kernels, possess antidiabetic, antiobesity, and anticancer effects, contributing to the preservation of the human cardiovascular system. The chemical composition of maize, especially in hybrid varieties studied in this research, is subject to variations influenced by genetic factors, agroclimatic conditions, and specific breeding practices. Understanding these factors is crucial for elucidating the nutritional potential and health benefits associated with consuming different maize hybrids.“ |
At the end of the introduction, it said that “The results obtained from this study provide insights into how agro-ecological conditions impact the nutritional potential of maize hybrids.”: but what were the agro-ecological conditions studied here? Results showed only differences in the chemical composition according to hybrid maize samples growing under the same agricultural practices. |
Following the referee's request, we have inserted agroecological conditions for this study in 2.1. Material section. “Serbia experiences a temperate continental climate, characterized by distinctive local features and a gradual shift between seasons. The maize crop yields are signifi-cantly influenced by weather factors, particularly precipitation and temperature pat-terns. Meteorological data, encompassing temperature and total precipitation, were gathered from the automated weather station at the Rimski Sancevi Agrometeorologi-cal Experimental Station [17]. Novi Sad (Rimski Sancevi), situated in northern Serbia at coordinates 45° 15′ N, 19° 50′ E, stands on the Pannonian Plain at an elevation of 80–86 meters above sea level.” |
What do the FAO maturity groups correspond to? Can you give a brief description and cite references. |
According to referee’s opinion we given a brief description FAO maturity groups in 2.1 material and methods. “All maize hybrids are classified into the international FAO maturity groups, depending on the length of the growing season (Table 1). The FAO group 100 is the earliest and 700 the latest maturity group. A higher number indicates a longer growing season and whether more heat units are required for the variety to reach grain maturity.” We have inserted table 1 entitled FAO maturity groups and the length of maize growing period. |
Table 2: remind FAO maturity group for studied hybrid maize sample or group them according FAO maturity group for memory. What is the range, the dispersion of chemical data in the same FAO maturity group? Give the minimum and the maximum values. |
Based on the referee's feedback, we have revised Table 2 according to FAO maturity groups. The hybrids are now grouped based on their FAO maturity group, and we have provided a range for all investigated parameters within the same FAO maturity group. The updated table can now be found as Table 3 in the results and discussion section. |
Same remarks for table 3. |
In response to the referee's feedback, Table 3 has been revised. The hybrids are now organized according to FAO maturity groups, and we have provided a range for all investigated parameters within each respective FAO maturity group. This updated presentation is now reflected as Table 4 in the results and discussion section. |
Why FA contents were not presented for each hybrid maize samples; only an average value of FA contents were presented according to FAO maturity groups…why? All maize oils obtained from hybrid samples of a same group have been mixed??? |
In table 4, statistical significance was obtained by ANOVA analysis according to FAO maturity groups, which was not the case in the statistical ANOVA of data of previous two tables. |
Lines 204-205: Cite reference about the benefits of the ratio of n-6 and n-3 acids. |
In response to the referee's feedback, we have incorporated additional references based on their observation. 18. Stojanović, Z. S.; Uletilović, D. D.; Kravić, S. Ž.; Kevrešan, Ž. S.; Grahovac, N. L.; Lončarević, I. S.; Đurović, A. D.; Marja-nović Jeromela, A. M. Comparative Study of the Nutritional and Chemical Composition of New Oil Rape, Safflower and Mustard Seed Varieties Developed and Grown in Serbia. Plants 2023, 12 (11), 2160. https://doi.org/10.3390/plants12112160. 46. World Health Organization, 1994. Fats and Oils in Human Nutrition: Report of a Joint Expert Consultation, Rome, 19-26 October 1993 (No. 57). Food & Agriculture Org. |
Lines 216-243: in my opinion, this part is unnecessary if a PCA has been carried out all chemical data. Figure 1 is sufficient in itself to serve as a support for determining correlated or anti-correlated variables. Add a color code according to the FAO maturity group. Revised the legend of variable by removing units of chemical data (for example eq quertecin, etc) and give a detailed legend of used acronyms. Why SFA, UFA, MUFA and PUFA were not considered as variables in PCA? Why use crude fat ? |
In alignment with the referee's recommendation, we have removed part in lines 216-243 together with table 5. We agree that only Fig.1 is sufficient. We've introduced a color-coded scheme corresponding to the FAO maturity groups, along with explanatory text, to enhance the interpretation of the new PCA biplot (Figure 1). Also, PCA analysis with SFA, UFA, MUFA and PUFA is done as well as rewrited whole paragraph which refers to Figures 1. Crude fat is mentioned in Table 2 with the basic chemical composition of maize hybrids across studied FAO maturity groups that's why the same term is used in Figure 1. |
Sample preparation: Were the maize samples dried before grinding? In Material and methods
|
In response to the referee's observation and as a precautionary measure to safeguard the seeds from potential adverse effects at higher moisture levels, the samples were subjected to drying. This was done to ensure that the moisture content was reduced to below 14% by the time of analysis for safety reasons. |
What were the soil cultivation practices applied? What were the climatic conditions during the growing of the maize samples? |
In response to the referee's request, we have included additional details in Section 2.1 (Materials) outlining the agricultural technology employed, along with a description of the climatic conditions prevalent during the cultivation of the maize samples. “Winter wheat (Triticum aestivum L.) was the previous crop. The selected plots underwent plowing in October to a depth of up to 30 cm, followed by seedbed preparation before sowing in March using heavy-duty cultivators (Multi-Tiller) at a depth of 15 cm. On 22 April 2021, the crop was sown using a Wintersteiger AG pneumatic precision seed drill at a depth of 5 cm. The plot dimensions were 5 × 2.8 m, featuring intra-row spacing of 22 cm and row spacing of 70 cm.Throughout both years, weed control was executed through conventional chemical methods. Pre-emergence application included a dose of 1.4 l ha-1 combined with 3.5 L ha-1 a mixture containing 375 g l-1 S-Metolachlor, 125 g l-1 Terbuthylazine and 37.5 g l-1 Mesotrione. Post-emergence application was also conducted. During the vegetation season, control of Sorghum halepense sp. and other narrow-leaved weeds was achieved by applying Nicosulfuron or Rimsulfuron at a rate of 50-60 g ha-1.” |
How many days after sowing, was the maize harvested? |
The harvest of the maize sample was done according to the differences of the FAO maturity groups, you can see in Table 1 under the heading FAO maturity groups and the length of the maize growing period (Table 1). |
Sincerely yours,
PhD Nada Grahovac, Senior Research Associate
Corresponding author: Nada Grahovac
Institute of Field and Vegetable Crops,
Maksima Gorkog 30, 21101 Novi Sad, Serbia
Tel: +381 21 4898 321; Fax: +381 21 4898 418
E-mail address: [email protected]
Resubmission Date
21 December 2023
